# Effect of Magnet Position on Tipping and Bodily Tooth Movement in Magnetic Force-Driven Orthodontics

**DOI:** 10.3390/ma13163588

**Published:** 2020-08-13

**Authors:** Yoshiki Ishida, Yukinori Kuwajima, Cliff Lee, Kaho Ogawa, John D. Da Silva, Shigemi Ishikawa-Nagai

**Affiliations:** 1Department of Oral Medicine, Immunity and Infection, Harvard School of Dental Medicine, 188 Longwood Avenue, Boston, MA 02115, USA; yoshiki_ishida@hsdm.harvard.edu (Y.I.); yukinori_kuwajima@hsdm.harvard.edu (Y.K.); kaho_ogawa@hsdm.harvard.edu (K.O.); 2Department of Orofacial Sciences, Division of Periodontology, University of California, San Francisco School of Dentistry, 513 Parnassus Ave, San Francisco, CA 94143, USA; cliff.lee.dmd@gmail.com; 3Department of Restorative Dentistry and Biomaterial Sciences, Harvard School of Dental Medicine, 188 Longwood Avenue, Boston, MA 02115, USA; john_dasilva@hsdm.harvard.edu

**Keywords:** magnets, CAD/CAM, superimposition, orthodontics, typodont, 3D printer, 3D scanner

## Abstract

The goal of our study is to launch magnetic force-driven orthodontics. This continuous study investigated the influence of magnet position on tipping and bodily tooth movement, using 3D printing technology and digital analysis. Orthodontic typodont models (TMs) for space-closure were 3D printed to mimic maxillary central incisors. Nd-Fe-B magnets were placed in the middle third (Model-M), and the cervical third (Model-C), of the tooth. TMs, before and after movement, were digitally scanned and superimposed. The 3D digital coordinates (X, Y, and Z axes), and rotations (yaw, pitch, and roll) of the tooth crown and root, were calculated and compared between the two magnet position settings. Model-M showed higher rates of movement, but more rotation than Model-C (*p* < 0.01). The root apex of Model-M moved in the opposite direction of the crown (*R* = −0.29), indicating tipping movement. In contrast, the crown and root apex moved in the same direction (*R* = 0.56) in Model-C, indicating bodily movement. These patterns were confirmed in a typodont model of a moderate crowding case. The results validated that modifying the magnet position increased the amount of bodily tooth movement, and decreased rotation/tipping in an ex vivo setting.

## 1. Introduction

Rare earth magnets, such as Sm-Co and Nd-Fe-B magnets, are the strongest type of permanent magnet and were introduced in the 1970s and 1980s [1,2,3]. They are the most common rare earth permanent magnets in use today and less costly to produce than Sm-Co alloys [3,4,5,6].

Rare earth magnets possess many qualities that are beneficial for orthodontic applications [1,4,7,8]. Rare earth magnets can generate strong magnetic forces in a small physical form, allowing them to be bonded to teeth. Magnetic forces follow the inverse square law, so orthodontic force strength does not decrease as teeth move closer together [1,8,9,10,11]. Magnetic forces can pass through the mucosa and bone [9], and patient compliance, as in the case of orthodontic elastics, is not required [3]. Because of the physical form, oral hygiene may be more easily performed than with conventional fixed orthodontics with auxiliaries, such as hooks, elastics, and spring coils. The first report of the use of magnetic force to move teeth was in 1977 when Kawata and Takeda [12] described a technique using Co-Cr-Fe alloy magnetic brackets for the closure of anterior interdental spaces [1,3,4]. Since then, rare earth magnets have been used in orthodontic treatment with limited clinical indications. Prasad et al. reported that Nd-Fe-B rare earth magnets were used for diastema closure in a clinical study in 2016 [4]. Many other studies and clinical reports have used Nd-Fe-B rare earth magnets in orthodontic treatment, such as in the forced eruption of impacted teeth, the correction of class II malocclusion, and with functional appliances [4,5,8,11,13,14,15].

From an esthetic standpoint, magnets may not be ideal due to the black or metallic color. The optically transparent ferromagnetic nanogranular films have been introduced [16].

The ultimate goal is the establishment of magnetic force-driven orthodontics. Previously, we completed an analysis of the efficacy of Nd-Fe-B magnetic forces using a three-dimensional digital analysis of tooth movement and rotation in an ex vivo typodont model [17]. We found that magnets can achieve tooth movement in a space-closing and space-gain model, and moderate crowding in an ex vivo setting [17]. However, more tipping movement and rotation were observed when using attractive forces. In orthodontics, bodily tooth movement is often preferred to tipping tooth movement, as tipping may destructively influence the periodontal tissues in the cervical area [18]. It was also reported that hyalinization occurs less frequently during bodily tooth movement than tipping movement [19]. In this study, we hypothesized that the magnet position influences the amount of tipping movement. This study aimed to clarify the influence of the position of magnet placement on tipping and bodily tooth movement and rotation, by means of 3D digital analysis.

## 2. Materials and Methods

### 2.1. Design and Fabrication of Typodont Model Used

We used the same methods of our previous study [17]. The design of the maxillary central incisors is shown in Figure 1a. A 3D printer (MiiCraft 125, MiiCraft Inc., Hsinchu, Taiwan) was used to fabricate the tooth models. The typodont boxes with 112 landmarks (4.0 mm × 1.5 mm pentagonal cones) were created (Figure 1b) [17].

Ni-plated cylindrical Nd-Fe-B N52 magnets (NeoMag Co., Ltd., Chiba, Japan) were used [17]. The magnets were bonded to the tooth 2 mm apart, as shown in Figure 1b. Two magnet positions were tested in this study. The magnet was placed in the middle third (Model-M), and in the cervical third (Model-C, Figure 1b), of the tooth crown. The teeth were then stabilized in the typodont box using paraffin wax (Paraffin wax, GC Co., Ltd., Tokyo, Japan) [17].

### 2.2. Experiment Flowchart

The experiment flowchart is shown in Figure 2. Prior to tooth movement, the typodont models were scanned using a 3D laser scanner (Ortho Insight, Motion View LLC., Chattanooga, TN, USA) [20]. Thirty typodonts were then divided into 10 groups. The tooth movement was initiated by immersing the typodonts in a water bath (Joan Lab Digital Thermostatic Water Bath, Ningbo Yinzhou Joan Lab Equipment Co., Ltd., Ningbo, China) at 55 °C from 5 min and up to 50 min. After the tooth movement, the models were stabilized in a cold-water bath at 5 °C for 30 min, and the models were dried and scanned [17]. The stereolithography (STL) files of pre-movement and post-movement were then imposed using 3D data inspection software (GOM inspect, GOM, Braunschweig, Germany) [17].

### 2.3. Analysis of the Movement and Rotation

#### 2.3.1. Measurement Points for Movement and Rotation

The 3D coordinates (X, Y, Z axes) of UR1, UL1, and the center of root apex (CRA) were obtained (Figure 3a,b) using 3D data inspection software (GOM inspect) [17].

#### 2.3.2. Calculation of Tooth Movement and Rotation

The amount of 3D movement of the center of gravity (CG) of each tooth crown (ACG), the speed of X-axis movement (mm/min), and the movement of the center of root apex (ACR) in each of the 10 time-series were calculated using the same formulas used in our previous study [17].

### 2.4. Application of the Magnetic Force-Driven Technique to the Moderate Crowding Case

The typodont model with a moderate crowding used in our previous study was employed in this experiment (Figure 4). The attraction magnet force and repulsion force were applied with the same manner of our previous study [17]. For Model-M, the magnets were placed in the middle third, and magnets were placed in the cervical third for Model-C. The desired arch form was established using a nickel-titanium archwire (0.012 inch Sentalloy, TOMY INTERNATIONAL INC., Tokyo, Japan). The 3D movement and rotation were analyzed in the same manner of our previous study [17].

### 2.5. Statistical Analysis

The average of movement and rotation in each of two groups and the 10 time-series were used for the statistical analysis (SPSS, IBM, Armonk, NY, USA). Two-way ANOVA was used to compare the amount of tooth and root movement in two factors: magnet position and the duration of orthodontic movement provided (*p* < 0.01). Student’s t-test was used to compare average amount of movement between two magnet-setting models in each of the 10 time points (*p* < 0.01). Pearson’s correlation coefficient was used to analyze the association between the amount of tooth movement and root movement in two models of magnet settings in three axes, X, Y, and Z. (*p* < 0.01).

## 3. Results

### 3.1. Crown Movement

The magnet position and duration were both significant factors in crown movement in all three axes (*p* < 0.01). The movement observed in the X-axis on both models was the largest, which was the intended movement. Model-C indicated less movement than Model-M in the X-axis (at 30, 40, and 45 min) and Z-axis (at 25, 30, and 40 min). The amount of movement on the X-axis in 50 min on Model-M and Model-C was 1.01 mm and 0.64 mm, respectively (Figure 5).

### 3.2. Speed of Movement

The magnet position influenced the rate of movement, with the maximum rate of movement on Model-M and Model-C at 0.050 mm/min and 0.043 mm/min, respectively (Figure 6). The greatest rate of movement was observed on the X-axis, which was the intended direction of movement.

### 3.3. Tooth Rotation

The magnet position and duration were both significant factors on yaw and roll (*p* < 0.01). The average degree of yaw at 50 min in Model-M (3.22) was significantly larger than that of Model-C (1.39, *p* < 0.01). When considering roll, significantly greater rotation in Model-M was observed than in Model-C in 50 min, at 2.89 degrees and 0.67 degrees, respectively (*p* < 0.01). However, there were no significant differences between Model-M and Model-C in the amount of pitch (Figure 7).

### 3.4. The Relationship between Tooth and Root Movement

The magnet position was a significant factor in the direction of root apex movement (Figure 8). In Model-M, ACR moved in the opposite direction of ACG, but in Model-C, ACG and ACR moved in the same direction.

In Model-M, a weak negative Pearson’s correlation coefficient was observed between ACG and ACR on the X-axis (*R* = −0.29), and a strong positive correlation was observed on the Z-axis (*R* = 0.92) (Figure 9a–c). In Model-C, a moderate positive correlation was found (*R* = 0.56) on the X-axis, and a strong positive correlation was observed on the Z-axis (*R* = 0.78) (Figure 9d–f).

### 3.5. Application of the Magnetic Force-Driven Technique to the Moderate Crowding Case

The different patterns were seen in the crown and root apex movement based on the magnet position (Figure 10). In Model-M, tipping movement was observed, in which the root apex moved in the opposite direction of the cusp. In contrast, in Model-C, bodily movement was observed, in which the cusp and root apex both moved distally. Less tooth rotation was observed in Model-C compared to Model-M.

## 4. Discussion

It was reported that optimal orthodontic treatment requires a mechanical input that leads to a maximum rate of tooth movement with minimal damage to the tooth root, periodontal ligament, and alveolar bone [21]. The amount of optimal force for orthodontic tooth movement has not yet been defined [22]. Higher forces do not always move teeth faster than lower forces, but higher forces result in more areas of hyalinization [23].

Recent studies have indicated that continuous forces result in faster tooth movement than intermittent forces [24,25,26], but continuous forces caused greater root resorption, with more unwanted rotational movement. Root-resorption and alveolar bone resorption are biological damage that has been observed with inappropriate orthodontic force application [24,25,26,27]. It is reported that unwanted rotational movement results in more root resorption in the middle third of the tooth [24]. Ideally, the magnitude of optimal force should be determined individually for each patient, and intermittent force should be applied to minimize unwanted outcomes by monitoring tooth movement closely. Accurate force control is an important factor.

Different types of loading force produce different types of tooth movement [28]. Cervical bone resorption and total alveolar bone thickness at the mid-root and apical levels were decreased with tipping movements compared to bodily movement [18,29]. Different settings of auxiliary appliances can create the different movements, such as use of the sliding tube, round-wire or square-wire, and shapes of advancing loops [18,29]. In our previous study, which used the same methods for digital scanning and measurements, attractive magnet force applied in the middle of maxillary central incisor for 2 mm diastema closure in an ex vivo model created unwanted tipping movement [17]. Therefore, we assessed the effects of two different positions of magnet placement, one at the middle third and the other in the cervical third, on the tipping movement and rotation.

As we hypothesized, the position of the magnet was a significant factor in creating tipping or bodily movement. Bodily movement is produced with linear force, but tipping forces are produced with lever force with a fulcrum, which in the case of orthodontics is the level of the alveolar bone, or in the case of this ex vivo study is the level of the paraffin wax. By moving the force more incisally, the distance of the lever force to the fulcrum increases, which for the same amount of force would produce a higher proportion of torque/rotation force. By moving the force more apically, the opposite occurs, decreasing the amount of tipping force and increasing the linear force for bodily movement. This was confirmed with the 3D digital analysis which indicated that Model-C created more bodily movement than Model-M. Although Model-M had a greater rate of movement due to the increased force output with a lever, it created more unwanted tipping/rotation. This result was similar to the report by Lee et al. [30]. The use of a greater magnetic force at the cervical area can compensate for the decrease of tooth movement distance and rate while minimizing unwanted rotation. The moderate crowding case validated that Model-C created bodily movement and less rotation on the canines with attracting force.

Placing orthodontic brackets in conventional fixed orthodontics at the cervical aspect of the crown creates unwanted side effects due to the proximity to the gingival margin, with gingival enlargement/hyperplasia and gingivitis being the most common effect [31]. This is likely due to a combination of difficult oral hygiene leading to plaque accumulation, as well as interactions between the metal from the brackets and the periodontium. Magnets could offer a significant advantage in this regard, with a much smaller profile without additional hooks or coiled springs, allowing for better oral hygiene and minimizing plaque accumulation.

The use of a typodont model is one of the limitations of this study [17]. One drawback to the ex vivo environment is the absence of oral fluids, which may potentially create corrosion of the magnets. However, corrosion-resistant magnets and new coating methods using nickel/alumina composite and multilayer titanium nitride ceramic, have been introduced to resist corrosion [32]. Typodont models have been used for dental education in many disciplines. They have also been a useful model for research in the field of orthodontics. Most recently, typodont models were used to test new bracket placement methods and new material for clear aligners [33,34,35]. Another study also used malocclusion typodont models to compare root movement measured by cone-beam computed tomography and consecutive digital scans [36]. The use of typodont models can be a great method in orthodontic research to mimic malocclusion, and test tooth movement prior to the consideration of clinical application. However, further research is required to study in the clinical environment with more complex and dynamic biological processes.

## 5. Conclusions

The 3D data obtained in this study indicated that magnet placement in the cervical area decreased undesired tipping/rotation and increased the amount of bodily movement. The magnets achieved desired bodily tooth movements in a moderate crowding ex vivo model with an attractive force.

## Figures and Tables

**Figure 1 materials-13-03588-f001:**
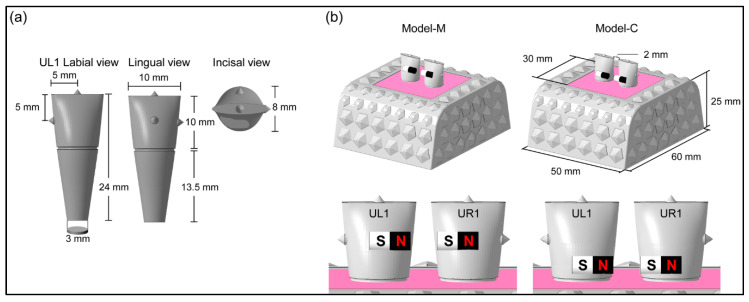
Typodont model used in this study. (**a**) Design and sizes of maxillary left central incisor (UL1), (**b**) scheme of typodont model.

**Figure 2 materials-13-03588-f002:**
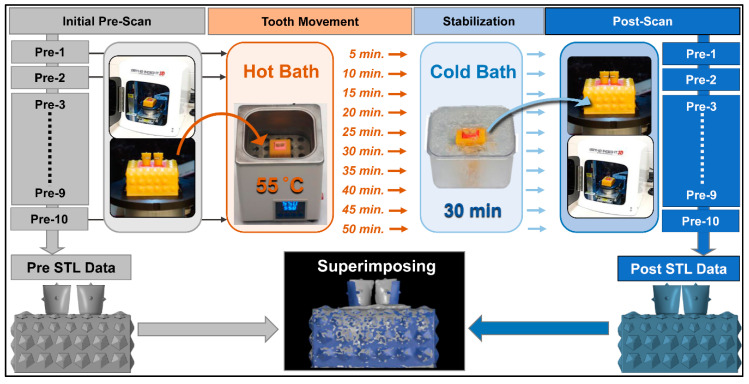
Experiment flowchart.

**Figure 3 materials-13-03588-f003:**
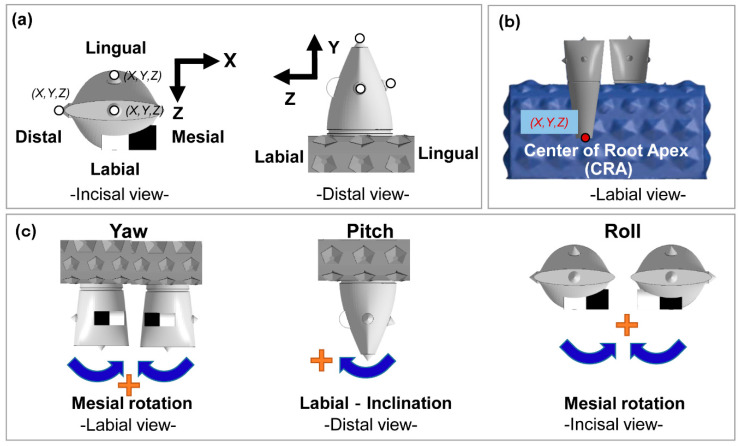
Scheme of tooth movements and rotations. (**a**) X, Y, and Z-axis direction of tooth movement, (**b**) identification of the center of root apex CRA). (**c**) rotations: yaw, pitch, and roll.

**Figure 4 materials-13-03588-f004:**
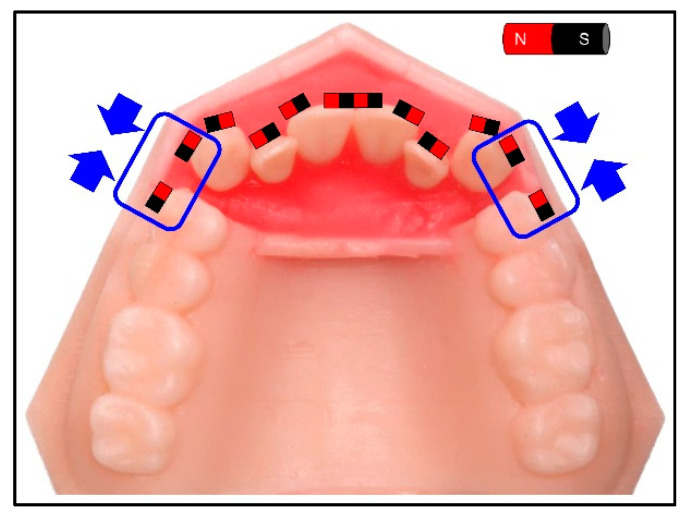
The magnet settings on the model of a moderate crowding case. Canines are supposed to move distally by magnetic force.

**Figure 5 materials-13-03588-f005:**
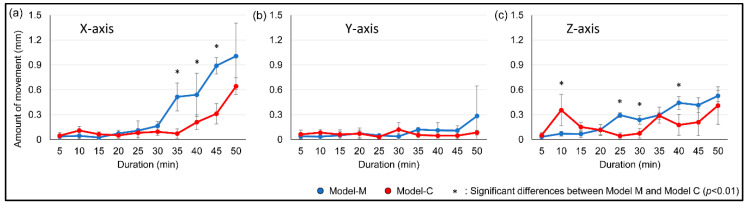
Average amount of tooth movement on the amount of 3D movement of the center of gravity (ACG). (**a**) X-axis, (**b**) Y-axis, (**c**) Z-axis.

**Figure 6 materials-13-03588-f006:**
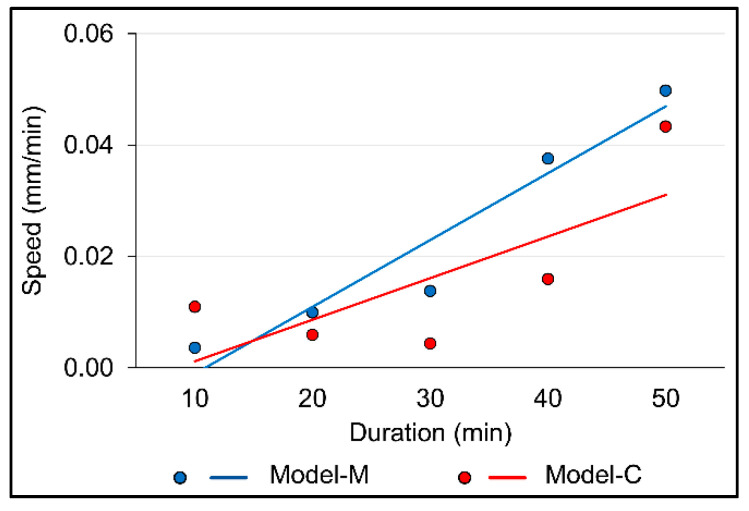
Average movement speed of the center of gravity.

**Figure 7 materials-13-03588-f007:**
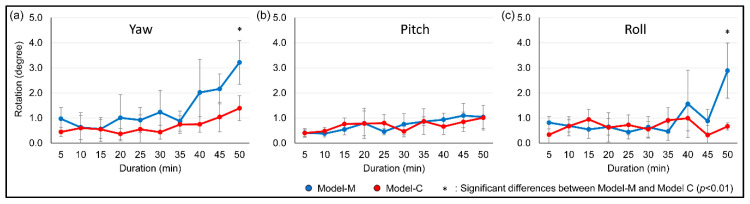
Average amount of rotation. (**a**) yaw, (**b**) pitch, (**c**) roll.

**Figure 8 materials-13-03588-f008:**
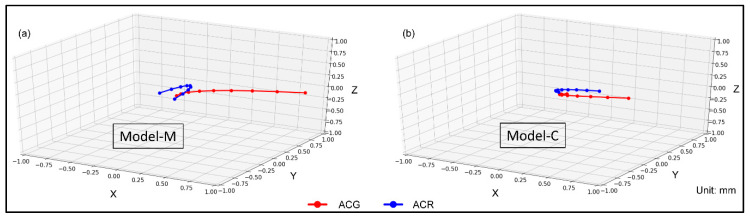
3D graphs of the amount of movement of the ACG and the amount of 3D movement of the center of root apex (ACR). (**a**) Model-M, (**b**) Model-C.

**Figure 9 materials-13-03588-f009:**
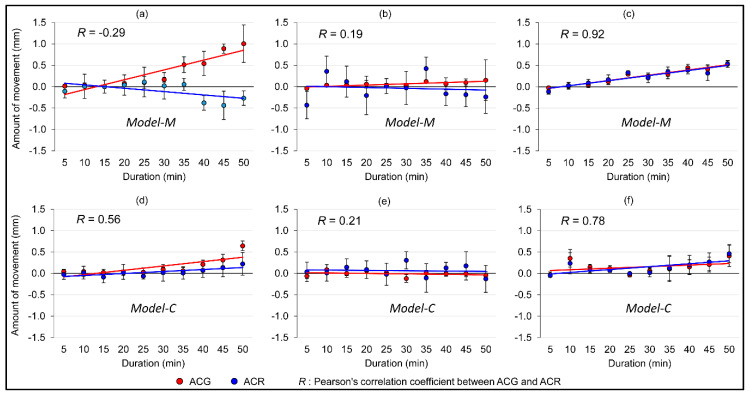
Average amount of 3D movement of the ACG and the ACR. (**a**) X-axis, (**b**) Y-axis, (**c**) Z-axis on Model-M, and (**d**) X-axis, (**e**) Y-axis, (**f**) Z-axis on Model-C.

**Figure 10 materials-13-03588-f010:**
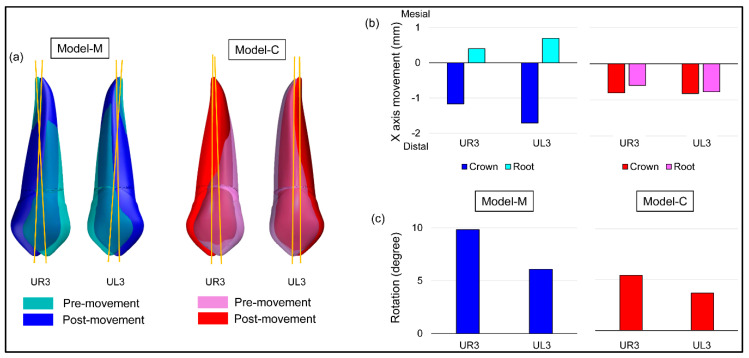
Results of tooth movement with magnetic force. (**a**) Images of maxillary right canine (UR3) and maxillary left canine (UL3) of pre- and post-movement, (**b**) distance moved on crown and root apex, (**c**) mesial tooth rotation.

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
