# Peer review of "Effect of Magnet Position on Tipping and Bodily Tooth Movement in Magnetic Force-Driven Orthodontics"

_materials, 2020, doi:10.3390/ma13163588_

Round 1

Reviewer 1 Report

“Effect of magnet position on tipping and bodily tooth 3 movement in magnetic force-driven orthodontics” is an interesting paper that investigated the effect of magnet position on tipping and bodily tooth movement using 3D printing technology and 3D digital analysis. Howevere some corrections are necessary before it can be considered sufficiently valid for publication.

First of all an extensive editing of English language and style is required, because the language is not very fluent and the translation is often too grammatical

Introduction

The characteristic of Rare earth magnets, their advantages for orthodontic applications, their clinical indications and limits need to be discussed in more detail.

Materials and Methods

This section is well described and supported by adequate iconography

Results

Results are described quite clearly and analytically.

Graphical representation is well done and allows a faster understanding of the results obtained in the study.

However, there is no adequate statistical analysis of the results obtained, which could validate what the authors claimed

An adequate statistical analysis of the results is necessary before the paper can be considered valid for pubblication

Discussion and Conclusions

The discussion and conclusions are well articulated overall, however they lack a statistical validation of the results achieved so they cannot be considered significant

Reviewer 2 Report

The study of “Effect of magnet position on tipping and bodily tooth movement in magnetic force-driven orthodontics” evaluate the orthodontic tooth movement driven by magnetic force. Additionally, two types of orthodontic tooth movement—included tipping and bodily movements by using two different magnet positions were estimated.

Overall, it is an interesting and clinical orientated study. However, some major problems are existed in the manuscript and needed to be clarify. In the section of Materials and Methods, the theoretical description of the experiment is clear; however, the real process of the experiment, especially how to measure the displacement through the machine, is unclear. The figures are displayed with the computer model of the STL, lacking pictures of the actual model and pictures related to the actual procedure of experiment. Readers need more information to confirm the credibility of the experiment.

Second, since the description of the actual experimental process is not very clear, what is the machine that measures the tooth movement? What is the resolution of the machine (especially the measurement of deformation)? How to determine that the measured difference of 0.01 per minute is correct?

Third, although the author mentioned the limitation of typodont in the discussion section, I am more concerned about the materials of typodont. Typodont is usually a denture model used for education, not for experiments. Is its material similar to the real oral condition? If the material is very different from the oral condition, how does the data of the movement of teeth can reflect the real situation?

Round 2

Reviewer 1 Report

"Effect of magnet position on tipping and bodily tooth movement in magnetic force-driven orthodontics" is an interesting paper which analyzes with good methodological rigor a fairly specialized biomechanical problem but in any case of good interest for the clinician.

The authors largely made the corrections required during the first review process, although some changes could be made more extensively, this version of the paper can be considered valid for publication.

Reviewer 2 Report

The authors have revised the manuscript based on reviewer’s comments. I have no more comments.